

**Using machine learning algorithm to retrieve cloud fraction based on**
**FY-4A AGRI observations**
Jinyi Xia[1]      Li Guan[1]
[1]China Meteorological Administration Aerosol-Cloud and Precipitation Key
Laboratory, Nanjing University of Information Science and Technology, Nanjing
210044, China
Correspondence to: Li Guan    liguan@nuist.edu.cn

9                                    **Abstract**

Cloud fraction as a vital component of meteorological satellite products plays an
essential role in environmental monitoring, disaster detection, climate analysis and
other research areas. A long short-term memory (LSTM) machine learning algorithm
is used in this paper to retrieve the cloud fraction of AGRI (Advanced Geosynchronous
Radiation Imager) onboard FY-4A satellite based on its full-disc level-1 radiance
observation. Correction has been made subsequently to the retrieved cloud fraction in
areas where solar glint occurs using a correction curve fitted with sun-glint angle as
weight. The algorithm includes two steps: the cloud detection is conducted firstly for
each AGRI field of view to identify whether it is clear sky, partial cloud or overcast
cloud coverage within the observation field. Then the cloud fraction is retrieved for the
scene identified as partly cloudy. The 2B-CLDCLASS-LIDAR cloud fraction product
from Cloudsat& CALIPSO active remote sensing satellite is employed as the truth to

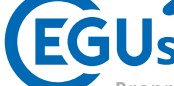

assess the accuracy of the retrieval algorithm. Comparison with the operational AGRI
level 2 cloud fraction product is also conducted at the same time. During daytime, the
probability of detection (POD) for clear sky, partly cloudy, and overcast scenes in the
official operational cloud detection product were 0.5359, 0.7041, and 0.7826,
respectively. The POD for cloud detection using the LSTM algorithm were 0.8294,
0.7223, and 0.8435. While the operational product often misclassified clear sky scenes
as cloudy, the LSTM algorithm improved the discrimination of clear sky scenes, albeit
with a higher false alarm rate compared to the operational product. For partly cloudy
scenes, the mean error (ME) and root-mean-square error (RMSE) of the operational
product were 0.2374 and 0.3269. The LSTM algorithm exhibited lower ME (0.1134)
and RMSE (0.1897) than the operational product. The large reflectance in the sun-glint
region resulted in significant cloud fraction retrieval errors using the LSTM algorithm.
However, after applying the correction, the accuracy of cloud cover retrieval in this
region greatly improved. During nighttime, the LSTM model demonstrated improved
POD for clear sky and partly cloudy scenes compared to the operational product, while
maintaining a similar POD value for overcast scenes and a lower false alarm rate. For
partly cloudy scenes at night, the operational product exhibited a positive mean error,
indicating an overestimation of cloud cover, whereas the LSTM model showed a
negative mean error, indicating an underestimation of cloud cover. The LSTM model
also exhibited a lower RMSE compared to the operational product.
**Key words:** Cloud detection, cloud fraction, FY-4A AGRI, LSTM neural network.





**Introduction**
Clouds occupy a significant proportion within satellite remote sensing data
acquired for Earth observation. According to the statistics from the International
Satellite Cloud Climatology Project (ISCCP), the annual average global cloud coverage
within satellite remote sensing data is around 66% with even higher cloud coverage in
specific regions (such as the tropics) (Zhang , et al., 2004). The impact of clouds on the
radiation balance of the Earth's atmospheric system is determined by the optical
properties of clouds. Cloud detection, as a vital component of remote sensing image
data processing, is considered a critical step for the subsequent identification, analysis,
and interpretation of remote sensing images. Therefore, accurately determining cloud
coverage is essential in various research domains, such as environmental monitoring,
disaster surveillance and climate analysis.
Fengyun-4A (FY-4A) is a comprehensive atmospheric observation satellite
launched by China in 2016. The uploaded AGRI (Advanced Geosynchronous Radiation
Imager) has 14 channels and captures full-disk observation every 15 minutes. In
addition to observing clouds, water vapor, vegetation and the Earth's surface, it also
possesses the capability to capture aerosols and snow. Moreover, it can clearly
distinguish different phases and particle size of clouds and obtain high- to mid-level
water vapor content. It is particularly suitable for cloud detection due to its
simultaneous use of visible, near-infrared and long-wave infrared channels for



observation with high spatial resolution.

Numerous cloud detection algorithms have been provided based on observations

from satellite-borne imagers. The threshold method has been widely employed by
researchers, encompassing the early ISCCP (International Satellite Cloud Climatology
Project) method (Rossow, 1993) and the proposed threshold methods based on different
spectral features or underlying surfaces. Kegelmeyer (1994) used a straightforward
cloud pixel as threshold for cloud detection with Whole Sky Imaging Cameras.
Solvsteen (1995) distinguished cold water pixels and cloud pixels by analyzing the
correlation between different channels based on AVHRR (Advanced Very High
Resolution Radiometer) images. A grouping threshold method based on AVHRR
images has been developed by Baum and Trepte (1996) to classify scenes as clouds,
fires, smoke or snow. LI and Zhang (2006) proposed a multispectral integrated cloud
detection algorithm based on the characteristics of MODIS instrument channels and the
spectral characteristics of different objects (clouds, snow, land, etc.). Zhang et al. (2020)
used a multi-temporal cloud detection method based on FY-4A AGRI data to identify
observations on the Qinghai-Tibet Plateau. However, there is a significant subjectivity
in selection of thresholds whether it is the single and fixed threshold in the early days,
multiple thresholds, dynamic thresholds, or adaptive thresholds. These thresholds are
highly influenced by factors such as season and climate.

The other category of cloud detection algorithms is the based on statistical

probability theory. Such as the principal component discriminant analysis and quadratic



discriminant analysis methods were used to SEVIRI (Spinning Enhanced Visible and
Infrared Imager) cloud detection (Amato et al., 2008). The cloud detection algorithm
for Thermal Infrared (TIR) sensor was based on the Bayesian theory of total probability
(Merchant et al., 2010) and the naive Bayes algorithm for AGRI (Qu , et al., 2022). The
unsupervised clustering cloud detection algorithms for MERIS (Medium Resolution
Imaging Spectrometer) (GomezChova , et al., 2007) and the fuzzy C-means clustering
algorithms for MODIS (Pan, et al., 2009) all have achieved high accuracy in cloud
detection.

More and more machine learning algorithms are being utilized by researchers in

cloud detection studies with the development of machine learning. For instance, the
probabilistic neural networks, especially radial basis function networks was used for
AVHRR cloud detection (Zhang, et al., 2001). The utilization of convolutional neural
network methods (Hu, et al., 2020) offers important perspectives for cloud detection
research.

Currently, there is limited research literature on cloud detection and cloud fraction

retrieval algorithms for FY-4A/4B AGRI. The operational cloud fraction product of FY-
4A AGRI utilized a threshold method with 4 km spatial resolution. Differences in
climatic and environmental factors lead to varying albedo and brightness temperature
observations for the instrument at different times and locations. Therefore, the choice
of thresholds is easily influenced by factors such as season, latitude and land surface
type (Gao and Jing, 2019). Using multiple sets of thresholds for discrimination would



significantly slow down the cloud detection process. Moreover, most algorithms focus
solely on cloud detection, which classified the observed scenes into cloud or clear-sky
without providing the specific cloud fraction information for the scenes.
In summary, a LSTM (Long Short-Term Memory) machine learning algorithm for
cloud fraction retrieval was established using level-1 radiation observations from FY-
4A AGRI full-disk scanning in this paper. The cloud fraction of the level-2 product 2B-
CLDCLASS-LIDAR from Cloudsat&CALIPSO was used as the reference label. The
retrievals were compared against with the cloud fraction of 2B-CLDCLASS-LIDAR
and the AGRI operational products to verify the algorithm accuracy.
**1 Research Data and Preprocessing**
*1.1 FY-4A data*
FY-4A was successfully launched on December 11, 2016. Starting from May 25, 2017,
FY-4A drifted to a position near the main business location of the Fengyun
geostationary satellite at 104.7 degrees east longitude on the equator. Its successful
launch marked the beginning of a new era for China's next-generation geostationary
meteorological satellites as an advanced comprehensive atmospheric observation
satellite. The Advanced Geosynchronous Radiation Imager (AGRI), one of the main
payloads of the Fengyun-4 series geostationary meteorological satellites, can perform
large-disk scans and rapid regional scans at a minute level. It has total 14 observation
channels with the main task of acquiring cloud images. The channel parameters and



main uses of AGRI are detailed in Table 1. FY-4A AGRI data was downloaded from
the official website of the China national satellite meteorological center
(http://satellite.nsmc.org.cn), including level-1 full disk radiation observation data
preprocessed through quality control, geolocation and radiation calibration as well as
level-2 cloud fraction product (CFR). The spatial resolution of these data is all 4 km
and the temporal resolution is 15 minutes.
**Table 1** FY-4A AGRI channel parameters

| Channel Number | Band Range /μm | Central Wavelength /μm | Spatial resolution/km | Main Applications |
|---|---|---|---|---|
| 1 | 0.45 ~ 0.49 | 0.47 | 1 | clouds, dust, aerosols |
| 2 | 0.55 ~ 0.75 | 0.65 | 0.5 | clouds, sand dust, snow |
| 3 | 0.75 ~ 0.90 | 0.825 | 1 | vegetation |
| 4 | 1.36 ~ 1.39 | 1.375 | 2 | cirrus |
| 5 | 1.58 ~ 1.64 | 1.61 | 2 | clouds、snow |
| 6 | 2.10 ~ 2.35 | 2.225 | 2 | cirrus、aerosols |
| 7 | 3.50 ~ 4.00 | 3.75H | 2 | fire point, the intense solar reflection signal |
| 8 | 3.50 ~ 4.00 | 3.75L | 4 | low clouds, fog |
| 9 | 5.80 ~ 6.70 | 6.25 | 4 | upper-level water vapor |
| 10 | 6.90 ~ 7.30 | 7.1 | 4 | mid-level water vapor |
| 11 | 8.00 ~ 9.00 | 8.5 | 4 | subsurface water vapor |
| 12 | 10.30 ~ 11.30 | 10.8 | 4 | surface and cloud-top temperatures |
| 13 | 11.5 0~ 12.50 | 12.0 | 4 | surface and cloud-top temperatures |
| 14 | 13.2 ~ 13.8 | 13.5 | 4 | cloud-top height |


## 1.2 CloudSat & Calipso Cloud Product

CALIPSO (Cloud-Aerosol Lidar and Infrared Pathfinder Satellite Observations)
is a satellite jointly launched by NASA and CNES (the French National Center for



Space Studies) in 2006. It is a member of the A-Train satellite observation system.
CALIPSO is equipped with three payloads, among which CALIOP (the Cloud and
Aerosol Lidar with Orthogonal Polarization) is a primary observational instrument.
Observing with dual wavelengths (532 nm and 1064 nm) CALIOP can provide high-
resolution vertical profiles of clouds and aerosols with 30 m vertical resolution. As the
first satellite designed to observe global cloud characteristics in a sun-synchronous orbit
CloudSat is also among NASA's A-Train series satellites. The CPR (Cloud Profile
Radar) installed on it operates at 94 GHz millimeter-wave and is capable of detecting
the vertical structure of clouds and providing vertical profiles of cloud parameters. The
scanning wavelengths of CPR and CALIOP are different. CALIOP is capable of
observing the top of mid-to-high level clouds, whereas CPR can penetrate optically
thick clouds. Combining the strengths of these two instruments enables the acquisition
of precise and detailed information on cloud layers and cloud fraction.

The joint level 2 product 2B-CLDCLASS-LIDAR is mainly utilizing in this study.

It provides the cloud fraction at different heights with horizontal resolution 2.5 km
(along-track) × 1.4 km (cross-track) through combining the observations from CPR and
CALIOP (Zhen, et al., 2018). The CloudSat product manual (Wang, 2019) can be
referred for more detailed information on 2B-CLDCLASS-LIDAR. The data used is
available for download from the ICARE data and services center
(https://www.icare.univ-lille.fr/data-access/data-archive-access/).



### 1.3  Establishment of Training Data


The crucial aspect of establishing a training data in machine learning algorithms
is how to obtain the cloud fraction values (ground truth) as labels. The error in cloud
fraction retrieved solely from passive remote sensing instruments is significant. Using
active remote sensing data can provide more accurate cloud fraction information in the
vertical direction. Therefore, the spatiotemporally matched 2B-CLDCLASS-LIDAR
cloud fraction are utilized as output labels in this paper.
The FY-4A AGRI and 2B-CLDCLASS-LIDAR data with a distance difference
between fields of view within 1.5 km and a time difference within 15 minutes are
spatiotemporal matched. To make the 2B-CLDCLASS-LIDAR cloud fraction data
collocated within AGRI pixels more effective, at least two 2B-CLDCLASS-LIDAR
pixels are required within each AGRI field of view. The cloud fraction average of these
pixels is used as the cloud fraction for that AGRI pixel.
Cloud detection and cloud fraction label generation for 2B-CLDCLASS-LIDAR
are as follows. There may be multiple layers of clouds in each field of view. If there is
at least one layer cloud with cloud fraction of 1 in the 2B-CLDCLASS-LIDAR profile,
then the scene is labeled as overcast with a cloud fraction of 1. If all layers in the profile
are cloud-free, the scene is labeled as clear sky. The scene between the above two
situations is labeled as partly cloudy and the cloud fraction is the average of cloud
fractions at different layers.



The algorithm includes two steps: the cloud detection is conducted firstly for each
AGRI field of view to identify whether it is clear sky, partial cloud or overcast cloud
coverage within the observation field. Then the cloud fraction is retrieved for the scene
identified as partly cloudy. So the training data include A dataset used for cloud
detection and B dataset for cloud fraction retrieval. The input variables in A dataset
are the FY-4A AGRI level-1 radiative observations from 14 channels and the output
variable is the temporally and spatially matched 2B-CLDCLASS-LIDAR cloud
detection label. The output is categorized into three types: overcast, partly cloudy and
clear sky with values 1, 2 and 3 respectively. To ensure diversity and representativeness
of the samples, the three conditions of overcast, partly cloudy, and clear sky each
account for one-third of the sample size in dataset A. Regarding the samples for partly
cloudy type in dataset A, the collocated 2B-CLDCLASS-LIDAR cloud fraction
products serve as output labels for cloud fraction retrieval model B. The input of
training dataset B remains the FY-4A AGRI level-1 radiative observations.
Due to the lifespan of the instrument only 2B-CLDCLASS-LIDAR data before
July 2019 can be obtained. So, the FY-4A AGRI observations and 2B-CLDLASS-
LIDAR matched in time and space in May 2019 are used as training samples to build
the algorithm model. The paired samples of whole June 2019 are served as the testing
samples to assess the model's retrieval accuracy. The number of training samples in
May are 12,420 for dataset A and 4140 for B. Testing samples in June are 15,459 for A
and 5,153 for B.





Although the retrieval model was trained and tested using 2019 data, the algorithm
was also applied to real-time observations of FY-4A and FY-4B AGRI in 2023 to verify
its universality.

**2.  Long Short-Term Memory (LSTM) Algorithm**
LSTM is an improved algorithm based on RNN (Recurrent Neural Network) with
the ability to retain long-term memory. and demonstrates improved performance in
longer sequences data comparing to ordinary RNNs (Sarker, 2001).   It can effectively
address the challenges of gradient explosion and gradient vanishing over time in
models., LSTM network has been extensively applied in diverse domains owing to its
distinctive features, such as meteorology and environmental prediction and so on (Bao,
et al., 2024; Bai and Shen. 2019). The structure of the LSTM unit is depicted in Figure
1. The update and transmission of historical information is facilitated through the
internal control of three states: the Forget Gate, the Input Gate and the Output Gate.
The pertinent mathematical expressions are:
$$f_t = \sigma(W_f^T \times [h_{t-1}, x_t] + b_f) \tag{1}$$
where $f_t$ denotes the output of the Forget Gate, $\sigma$ signifies the Sigmoid
activation function; $W_f^T$ and $b_f$ correspond to the weight and bias of the Forget Gate,
respectively, $x_t$ stands for the current input, $h_{t-1}$ represents the output from the



previous time step.
$$i_t = \sigma(W_i^T \times [h_{t-1}, x_t] + b_i) \tag{2}$$
where $i_t$ represents the information updated after $\sigma$ activation, $W_i^T$ and $b_i$
denote the weight and bias, respectively.
$$\widehat{C}_t = \sigma(W_c^T \times [h_{t-1}, x_t] + b_c) \tag{3}$$
$\widehat{C}_t$ signifies the information updated after tanh activation, $W_c^T$ and $b_c$ denote
the weight and bias, respectively.
$$C_t = f_t \times C_{t-1} + i_t \times \widehat{C}_t \tag{4}$$
$C_t$ is the current information of the LSTM structure, $C_{t-1}$ denotes the
information of the LSTM structure from the previous time step.
$$O_t = \sigma(W_O^T \times [h_{t-1}, x_t] + b_O) \tag{5}$$
$O_t$ is the current output information, $W_O^T$ and $b_O$ denote the weight and bias,
respectively.
$$h_t = o_t \times \tanh(C_t) \tag{6}$$
$h_t$ denotes the current output result.



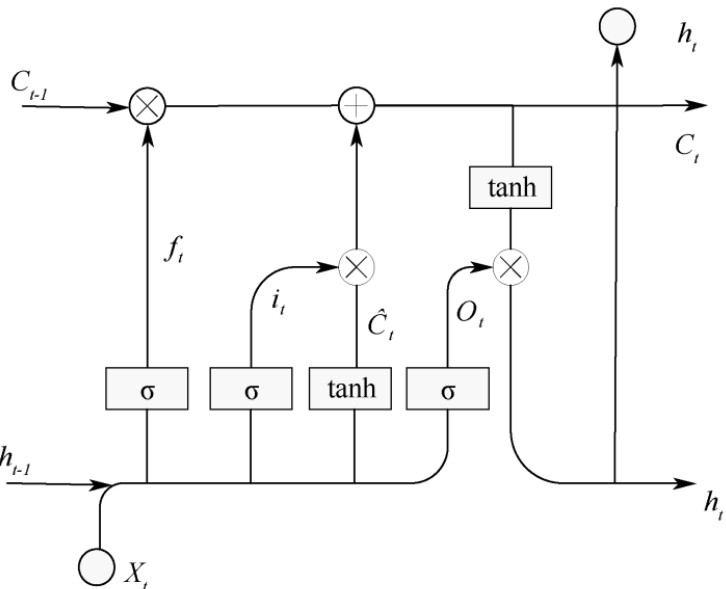


**Figure 1** LSTM cell structure (Kong, et al., 2018)

In a neural network, the hidden layer is a layer or multiple layers located between

the input layer and the output layer. Each hidden layer consists of multiple nodes, which
process the input data and generate outputs through connection weights and activation
functions. Increasing the size of the hidden layer can enhance the network's
representational capacity and learning ability, as more nodes can capture additional data
patterns and features. However, having a hidden layer that is too large may lead to
overfitting, making the network overly complex and difficult to train. Typically, the
optimal size of the hidden layer is determined by trying different sizes and evaluating
their performance on a validation set. The hidden layer sizes for both the cloud
classification model and the cloud fraction retrieval model in this paper are set to 3.

The key model parameter 'batch size' has two main impacts on training network:





(1) A larger batch size typically reduces the training time per epoch as more samples
are processed with each parameter update. On the contrary, a smaller batch size may
slow down the training speed since more iterations are needed to complete an epoch.
(2) Model Performance: Different batch sizes can impact the model performance.
Generally, a larger batch size may lead to quicker model convergence, yet it could
increase the risk of overfitting at times; whereas a smaller batch size could aid in the
model's generalization ability but might result in a less stable training process. In this
paper, the batch size of the model is set to 500.The optimizer is configured with the
Adam gradient descent algorithm, and the loss function used is cross-entropy.

The training dataset A was used to construct the LSTM cloud detection model. For

daytime, the inputs are the radiation observations from 14 channels of FY-4A AGRI
with 'input size' 14. However, during nighttime, as there are no observations in the
visible light channels (channels 1 to 6) of AGRI, the inputs consisted of the radiance
observations of channels 7 to 14 of FY-4A AGRI with 'input size' 8. The output label
is the classification of field of view, including overcast, partly cloudy and clear sky.

To derive the specific cloud fraction for AGRI scenes identified as partly cloudy

in the previous cloud mask step, an LSTM cloud fraction retrieval model needs to be
constructed. The training dataset B was used to train the cloud fraction retrieval model.
For daytime, the input is the observed radiances for all channels of AGRI (input
size=14), while during nighttime, the input comprises the observed radiance values of
channels 7 to 14 of AGRI (input size = 8). The output label is the value of cloud fraction



in the scene ranging from 0 to 1. When selecting parameters for the LSTM cloud
fraction model, a batch size of 60 is chosen due to the limited sample number in dataset
B. The optimizer is also configured with the Adam gradient descent algorithm. The loss
function used is mean square error.
**3.  Results and Analysis**

To assess the accuracy and stability of the retrieval model, two types of validation

methods are utilized. One way involves a direct comparison from images, qualitatively
comparing the model's retrieval results and official cloud fraction products with AGRI
observed cloud images. Another way is quantitative comparison using 2B-
CLDCLASS-LIDAR as the true value. Four quantitative parameters, including
possibility of detection(POD), alse alarm rate(FAR), mean error (ME) and root mean
square error (RMSE) are introduced. 'Possibility of detection' is calculated using the
formula POD=TP/(TP+FN), and false alarm rate is calculated using the formula
FAR=FP/(TP+FP). Taking the covercast scenes as an example, TP represents the
number of correctly identified overcast, FN represents the number of overcast scenes
wrongly identified as partly cloudy or clear sky, and FP represents the number of clear
sky or partly cloudy scenes wrongly identified as overcast.The ME (mean error) and
RMSE (root mean square error) are utilized to assess the accuracy of the LSTM cloud
fraction model in retrieving cloud fraction for partly cloudy scenes.



**3.1  Objective Analysis of Cloud Fraction Retrievals**

The test samples from dataset A (i.e., June data) are used to perform cloud

detection experiments based on the cloud detection model mentioned above. The
temporally and spatially matched 2B CLDCLASS-LIDAR cloud mask products are
used as reference to evaluate the accuracy of cloud detection. The POD and FAR for
different view field classifications are shown in Table 2. Columns 2 and 4 represent the
operational cloud detection products for daytime and nighttime respectively, for the
same time and pixel. Columns 3 and 5 represent the LSTM cloud detection results for
daytime and nighttime respectively. The table indicates that during daytime, operational
cloud detection products have a relatively low possibility of detection for clear sky view
fields. However, the LSTM model increases the possibility of detection for clear sky
from 0.54 to 0.83. Moreover, for some partly cloudy and overcast view fields, the
possibilities of detection is higher than those of operational cloud detection products.
During nighttime, compared to operational cloud detection products, the LSTM model
increases the POD for clear sky from 0.51 to 0.73, with slightly higher possibilities of
detection for partial cloud view fields than the operational products, while the
possibility of detection for full cloud view fields is lower. During the day, the
Operational product has a lower false alarm rate for clear sky compared to the LSTM
model, while the LSTM model has a lower false alarm rate for partly cloudy and
overcast conditions than the Operational product. At night, the LSTM model



significantly reduces the false alarm rate for overcast conditions compared to the
Operational product.
**Table 2** POD and FAR of Cloud Detection

|  | Sky Classification | Daytime Operational Cloud Detection Product | Daytime LSTM Results | Nighttime Operational Cloud Detection Product | Nighttime LSTM Results |
|---|---|---|---|---|---|
| POD | Clear Sky | 0. 5359 | 0.8294 | 0.5136 | 0.7341 |
|  | Partly cloudy | 0.7041 | 0.7223 | 0.6957 | 0.7101 |
|  | Overcast | 0.7826 | 0.8435 | 0.7984 | 0.7523 |
| FAR | Clear Sky | 0.2174 | 0.3633 | 0.1789 | 0.1983 |
|  | Partly cloudy | 0.2959 | 0.1677 | 0.3107 | 0.3488 |
|  | Overcast | 0.4641 | 0.2358 | 0.5543 | 0.2105 |


For the view fields judged as partly cloudy by the aforementioned model, the cloud
amount in the AGRI view field was inverted using the LSTM cloud amount model
established earlier in this text. For samples classified as partly cloudy by the model,
operational products and 2B-CLDCLASS-LIDAR cloud amount products, the mean
error and root mean square error (RMSE) of the cloud amount retrieval were calculated
based on the matched 2B-CLDCLASS-LIDAR cloud amount product as ground truth,
separately for daytime and nighttime operational cloud amount products (columns 2
and 4) and the LSTM-inverted cloud amount (columns 3 and 5), as shown in Table 3.
It can be observed that during daytime, compared to the FY-4A operational cloud
amount product, the LSTM cloud amount retrieval model shows significant
improvement in both mean error (ME) and RMSE. The ME decreases from 0.23 to 0.11,



and the RMSE decreases from 0.32 to 0.19, indicating that the LSTM cloud amount
retrieval model provides more accurate estimates of cloud amount. For nighttime, the
ME of the operational cloud amount product is positive, indicating an overall
overestimation of cloud amount. In contrast, the ME of the LSTM model is negative,
indicating an overall underestimation of cloud amount. The RMSE of the LSTM model
retrieval results during nighttime is lower than that of the operational cloud amount
product.
**Table 3** Errors in cloud fraction retrieval

|  | Daytime Operational Cloud Detection Product | Daytime LSTM Results | Nighttime Operational Cloud Detection Product | Nighttime LSTM Results |
|---|---|---|---|---|
| ME | 0.2374 | 0.1134 | 0.2488 | -0.1911 |
| RMSE | 0.3269 | 0.1897 | 0.3374 | 0.2361 |

**3.2 Cloud fraction correction in sun glint regions**
Sun glint refers to the bright areas created by the reflection of sunlight to the
sensors of observation systems (satellites or aircrafts). This phenomenon usually occurs
on extensive water surfaces, such as oceans lakes or rivers. This specular reflection of
sunlight will cause an increase in the reflected solar radiation received by onboard
sensors, manifested as an enhancement of white brightness in visible images. The
increase in visible channel observation albedo will affect various subsequent
applications of data, including cloud detection and cloud cover retrieval, etc.



The position of Sun glint area can be determined using the SunGlintAngle value
in the FY-4A GEO file. SunGlintAngle is defined as the angle between the satellite
observation direction or reflected radiation direction and the mirror reflection direction
on a calm surface (horizontal plane). It is generally accepted that the range of
SunGlintAngle < 15° is easily affected by sun glint (Kay S, et al., 2009). The positions
of the SunGlintAngle contour lines at 5 and 15° are marked in Figure 2(a). It can be
observed that the edge of sun glint in Figure 2(a) essentially overlaps with the position
of SunGlintAngle = 15°. Thus, the region where SunGlintAngle < 15° is defined as the
sun glint range in this paper and only the cloud fraction within this range will be
adjusted in the subsequent correction.
To correct the cloud fraction in the sun glint region, we initially identified 672
fields of view where sun glint occurred in the FY-4A AGRI observations between 1
June and 31 July 2019.   Subsequently, a direct least squares fitting was conducted
between the inverted cloud fraction and the collocated 2B-CLDCLASS-LIDAR cloud
fraction (ground truth). The scatter plot is illustrated in Figure 2(b), where x-axis is the
2B-CLDCLASS-LIDAR cloud fraction and y-axis is the model-inverted cloud fraction.
The blue line represents the curve (namely Eq.7) fitted by the least squares method
between the retrievals and the truths. The thin dash line is the x=y line. It is evident that
the inverted cloud fraction is generally slightly overestimated.
Taking observations at 04:00 on 5 June 2019 as an example, Figure 2(c) presents
the distribution of SunGlintAngle and the flight trajectory of the Cloudsat&Calypso



satellite. White circles denote the sun glint region with SunGlintAngle < 15° and the
white line represents the satellite flight track. As depicted in the figure, the majority of
Cloudsat&Calypso flight trajectories do not pass through the central position of sun
glint area but instead traverse locations with larger SunGliantAngle values. The
intensity of sun glint effect decreases with the increase of SunGliantAngle. This
suggests that the true values for spatial and temporal matching mostly do not fall within
the strongest sun glint region. From Figure 2(d), it can be seen that the impact of sun
glint becomes stronger as SunGlintAngle decreasing, which results in a higher
observation albedo. This further leads to the overestimated cloud fraction values in the
retrieval. It is evident that the cloud fraction error is related to the value of
SunGlintAngle and this influence is not considered in Eq. (7). Directly applying
equation (7) to correct the cloud fraction retrievals would result in a too small correction
intensity for the FOVs near the center of sun glint and an excessively large correction
intensity for the FOVs in the Sun-glint edge region (even erroneous clear sky may
appear). Considering this, a correction formula (8)-(9) using SunGlintAngle as weight
is introduced, where $W_i$ represents the angle weight for a certain pixel $i$ in the sun glint
region, n is the number of pixels within the SunGlintAngle < 15° range, yi is the initial
model retrieval of cloud cover for the field of view $i$ and $x_i$ is the final corrected cloud
fraction.

$x = (y - 0.2562)/0.8428$                             (7)




$$W_i = \frac{glintangle_i}{\frac{1}{n}\Sigma_{i=0}^{n} glintangle_i} \tag{8}$$
$$x_i = W_i \left(\frac{y_i - 0.2526}{0.8428}\right) \tag{9}$$
Figure 2(d) shows the distribution of errors with respect to SunGlintAngle,
where the blue dots represent the error distribution corrected using formula
(7), and the orange dots represent the error distribution corrected using
formula (9). It can be seen from Figure 2(d) that after correction by formula
(9), the errors in the smaller range of SunGlintAngle are significantly reduced.

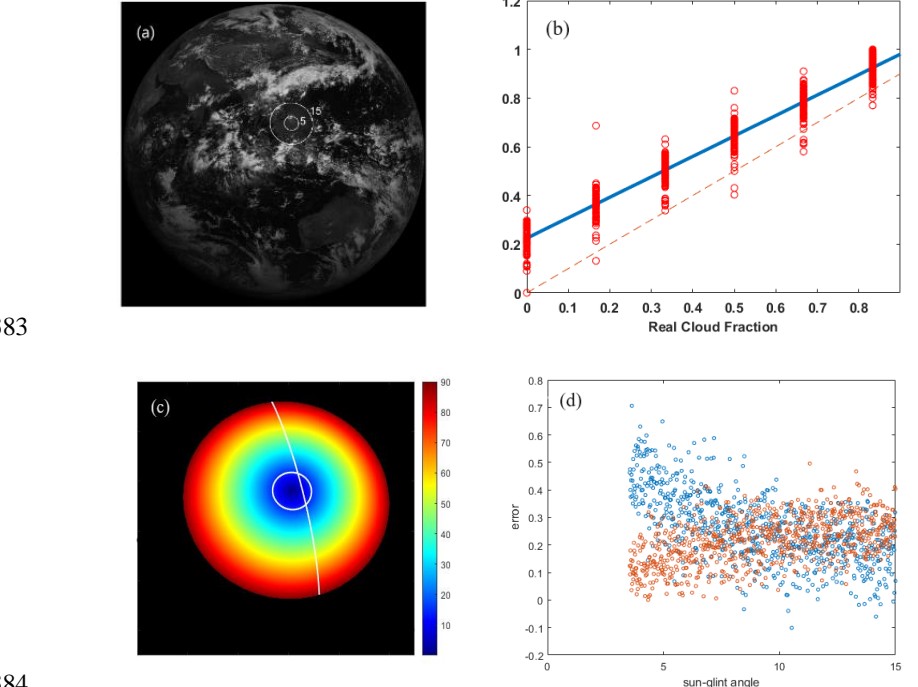


**Figure 2** (a) albedo image of 0.67μm channel (the circles are the contours of the sun-
glint angle), (b) Scatter plot of cloud fraction in sun glint region, (c) Distribution of
SunGlintAngle and satellite flight track of CloudSat & Calypso at 4:00 on June 5, 2019,





(d) Distribution of cloud fraction retrieval error with sun-glint angle.
**3.3 Algorithm universal applicability testing**
Although the retrieval model in this article was built based on data from May 2019
due to the limited lifespan of the instrument, how effective is it in real-time FY-4A
AGRI observations and even subsequent FY-4B AGRI applications? The algorithm's
universal applicability was tested using real-time observations from FY-4A and FY-4B
AGRI in 2023.
Taking the full-disk observation of FY-4A AGRI at 04:00 (UTC, the same below)
on 1 June 2023 as an example, the radiance observations from 14 channels are initially
fed into the LSTM cloud detection model to determine the sky classification (overcast,
partly cloudy or clear sky) in each AGRI field. The LSTM cloud fraction retrieval
model is utilized to estimate the cloud fraction in scenes identified as partly cloudy.
Figure 3(a) is the observed albedo at 0.67 μm, where the circles represent the contours
of the sunglint angle, (b) is the cloud fraction retrievals from LSTM algorithm, (c) is
the official operational cloud fraction product and (d) is LSTM cloud fraction retrievals
with sun-glint correction. It can be seen from Figure 3 that many clear-sky scenes are
erroneously identified as cloudy by the operational product and the cloud fraction is
generally overestimated with many scenes having a cloud fraction of 1. The LSTM
algorithm identifies more regions as clear skies or partly cloudy than the operational
products, matching better with the observations in the 0.67 μm albedo image. Brighter



regions in the visible image correspond to cloud cover areas and darker areas represent
clear sky conditions. The sun glint region in the central South China Sea (the circled
area in Figure 3(a)) is depicted in Figure 3(b), where the clear-sky scenes over the ocean
are misidentified as partly cloudy by LSTM algorithm due to the increase in observed
albedo. Although operational product in this area also suffers from the impact of
unremoved sun glint, it identifies more clear-sky scenes and the cloud fraction is
relatively low. Thus, it is evident that the LSTM algorithm exhibits significant cloud
detection and cloud fraction errors in these sun glint regions. Correction is necessary
for the cloud fraction retrievals in the sun glint region.

Figure 3(d) shows the cloud fraction distribution after correction using equation

(9) in the sun glint region., The correction eliminates the influence of sun glint
comparing to the cloud fraction in sun glint area before correction in Figure 3(b). The
scenes misjudged as partly cloudy are corrected to clear sky and match well with the
actual albedo observations in 3(a), which accurately restores the true cloud coverage
over the South China Sea.



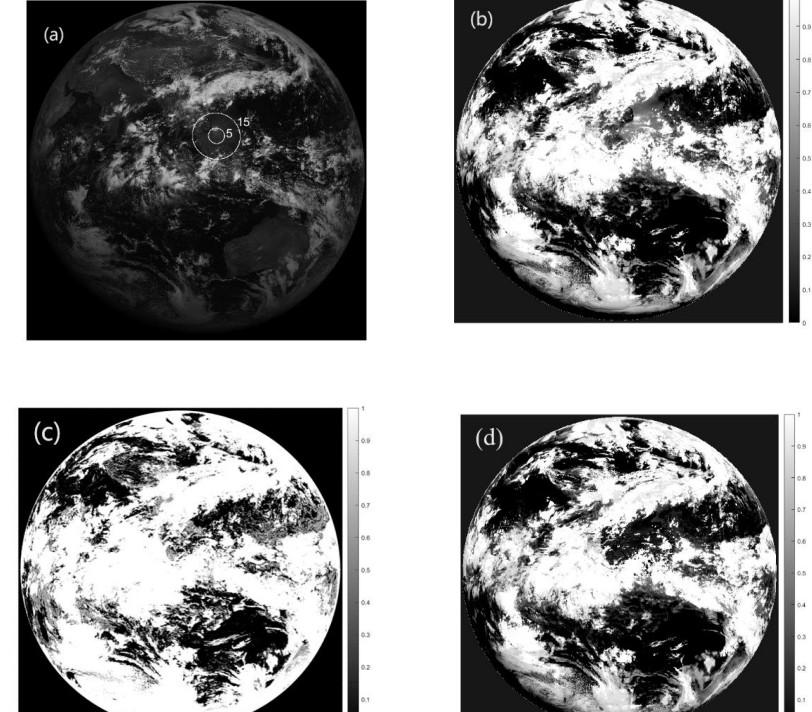



**Figure 3** FY-4A AGRI at 04:00 on 1 June 2023 (a) albedo image of 0.67μm channel
(the circles are the contours of the sun-glint angle), (b) LSTM cloud fraction
retrieval without sun-glint correction, (c) operational cloud fraction product, (d)
LSTM cloud fraction retrieval with sun-glint correction.
Statistical analysis was conducted on the correction effect using samples with sun
glint in the training data. The possibility of detection and false alarm rate in sun glint
area is listed in table 4 and the error is in table 5. The possibility of detection for clear
skies has increased from 0.09 to 0.83. The false alarm rate for partly cloudy has
decreased from 0.89 to 0.17. The mean error of cloud fraction retrievals decreased from





0.176 to 0.09. These all indicate that the positive effect of the sun glint correction.
**Table 4** The cloud mask recall rate in sun glint area

|     | Sky Classification | Operational Product | LSTM | LSTM after Correction |
|-----|-----|-----|-----|-----|
|     | Clear Sky | 0.5535 | 0.0900 | 0.8301 |
| POD | Partly cloudy | 0.6738 | 0.8279 | 0.7436 |
|     | Overcast | 0.8505 | 0.9744 | 0.9744 |
|     | Clear Sky | 0.1437 | 0.0063 | 0.3142 |
| FAR | Partly cloudy | 0.3742 | 0.8972 | 0.1719 |
|     | Overcast | 0.5545 | 0.1324 | 0.1324 |


**Table 5** cloud fraction Errors in sun glint area

|      | Operational Product | LSTM Retrievals | LSTM after Correction |
|------|-----|-----|-----|
| ME   | 0.2691 | 0.2760 | 0.1634 |
| RMSE | 0.3458 | 0.1948 | 0.1883 |

FY-4B launched in 2021 has a total of 15 channels with an additional low-level
water vapor channel at 7.42 μm compared to FY-4A. Taking the full-disk observation
of FY-4B AGRI at 17:00 on April 18, 2023, as an example, The radiance observation
data of the remaining eight channels (near-infrared and infrared channels) except for
the 7.42 μm channel and the visible light channels were input into the LSTM cloud
detection model. Figure 4 (a) shows the brightness temperature distribution observed



in the 10.8 μm channel of FY-4B AGRI, (b) represents the operational cloud fraction
product for FY-4B AGRI and (c) shows the cloud fraction retrieved by this algorithm.
Figure 4 illustrates that the LSTM algorithm identifies more regions as clear skies or
partly cloudy than the operational products, aligning better with the brightness
temperature observations in 10.8 μm. Especially in high latitude regions of the southern
hemisphere and areas with strong convection near the equator, the cloud cover provided
by operational products is too high and even misjudged. It can be seen that the LSTM
algorithm is also suitable for cloud fraction retrieval of FY-4B AGRI.

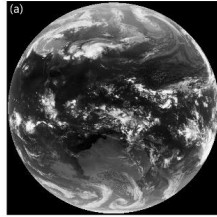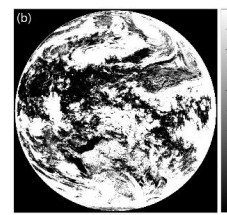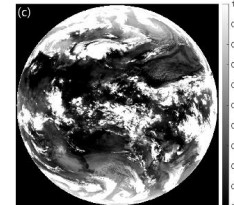


**Figure 4**    FY-4B AGRI at 17:00 on 18 April 2023, (a) brightness temperature of
10.8μm channel, (b) operational cloud fraction product, (c) LSTM cloud fraction
retrieval.

**4    Conclusion**

The long short-term memory (LSTM) machine learning algorithm based on FY-

4A AGRI full-disc level-1 radiance observations is developed to retrieve the cloud



fraction for each field of view in this paper. The accuracy of the algorithm is validated
using the 2B CLDCLASS-LIDAR cloud fraction product from the Cloudsat&Calypso
active remote sensing satellite and FY-4A AGRI level 2 operational product. The
following conclusions are drawn:
(1) Not only the cloud detection but also the cloud fraction within each FY-4A
AGRI field of view can be retrieved by the LSTM machine learning algorithm.
(2) The operational product has a relatively high false alarm rate for clear sky
scenes, while the LSTM algorithm improves the probability of detection (POD)
for clear sky scenes during the daytime from 0.54 to 0.83. However, the false
alarm rate (FAR) is higher compared to the operational product. The POD for
clear sky scenes at night increases from 0.51 to 0.73, and the POD for partially
cloudy and fully cloudy scenes is comparable to the operational product.
(3) For partly cloudy fields, during the day, the mean error and root-mean-square
error of the operational product are 0.2374 and 0.3269, respectively, while this
algorithm exhibits lower mean error (0.1134) and RMSE (0.1897) than the
operational product. At night, the operational product tends to overestimate
cloud cover, while this algorithm underestimates cloud cover, with a lower
RMSE compared to the operational product.
(4) The cloud fraction correction curve for sun glint region fitted with
SunGlintAngle as weight significantly improves the accuracy of the LSTM
cloud fraction retrievals. It reduces the misjudgment rate where increased



albedo leads to the identification of clear-sky scene as partly cloudy or overcast.

*Data availability*
FY-4A AGRI data is available at http://satellite.nsmc.org.cn and the 2B-CLDCLASS-
LIDAR data at https://www.icare.univ-lille.fr/data-access/data-archive-access/

*Author contributions*
JX: Formal analysis, Methodology, Software, Visualization and Writing – original draft
preparation. LG: Conceptualization, Data curation, Funding acquisition, Supervision,
Validation and Writing – review & editing.

*Competing interests*
he contact author has declared that none of the authors has any competing interests.

*Disclaimer*
*Acknowledgements*
Funding: This work was supported by the National Natural Science Foundation of
China under grant no. 41975028.

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
