# Peer review of "Retrieval of Cloud Fraction using Machine Learning Algorithms"

_EGUsphere, 2024_

## Author Response (AR1)

Refereer1 comments:

1-7. These are minor modifications.
Answer: Corrected as requested.

8. Line 241, 'hidden layer size' can be confusing for readers whether that is the number of hidden layers, or the number of neurons on each hidden layer. Personally I would believe the number 3 refers to the number of hidden layers. But the saying on Line 238 which reads as 'having a hidden layer that is too large' also seems to indicate the latter meaning. Please explain in a clear way.
Answer: Following the suggestion of referee2, LSTM is replaced with random forest. Random forest does not have the hyperparameter 'hidden layers'.

9. Line 251~252, are both models A and B using the same settings of batch size, optimizer and loss function? As far as I know, classification models (model A) and regression models (model B) usually use different loss functions.
Answer: As per Comment 8, LSTM has been swapped with random forest.

10-15. Substitution of words and modification of the title in Figure 2 (b).
Answer: Corrected as requested.

16. Table 4, what truth value are the statistics calculated against? According to Line 190~191 in this manuscript, 2B-CLDCLASS-LIDAR data is only available before July 2019.
Answer: Table 4 presents the statistical truth of 2B-CLDCLASS-LIDAR data for June and July. 2B-CLDCLASS-LIDAR data is available until August 2019. This error has been rectified.
Line 216.

Refereer2 comments:

Why do you only use data from May and June 2019? FY-4A was launched earlier, so shouldn't there be more data available? It would be good to take a random sample from a larger window of time with more seasonal variability. If there is some other reason only to use this time window, it needs to be stated here.
Answer: The latitude range for a single observation of FY-4A AGRI is -83.3~83.3. This latitude range includes data from different seasons, climates, and surface types. In the training samples matched temporally and spatially with 2B-CLDCLASS-LIDAR, seasons and climates vary with latitude. Therefore, it is not necessary to include data from a larger time range as training samples.
Line 215-220.

Figure 2:

(a) circular lines should be thicker, text should be larger

(b) missing a legend for the colors

(c) put the degree sign ° on the tick labels in the color bar

(d) missing a legend for the colors, and you should use different marker types (e.g. triangles and circles) to help colorblind readers

Answer: The figure has been redrawn as requested.

Line 381-382

You use a temporal / recurrent network architecture (LSTM) but have no description of how the temporal aspect of your data is used by the network. You need to describe this, in detail. What are the timesteps? Is there a warmup time for your LSTM (a minimum time before it attains a stable state and a reasonable accuracy)? On the other hand, if there is no time series information in use here, you should not be using an LSTM! Having a reasonable standard baseline (like a regular neural network or a random forest) would further help contextualize your results.

Answer: There was no use of time series information in this paper. Following the suggestion, LSTM has been replaced with Random Forest.

You are comparing your network with an operational cloud product. This is not a very fair comparison, as the operational cloud product is designed with different goals and principles (and physics) in mind than 2B-CLDCLASS-LIDAR was, whereas your method is directly trained on the 2B-CLDCLASS-LIDAR product. You should spend more time breaking down these differences.

Answer:    Additional algorithms for the 2B-CLDCLASS-LIDAR product were provided(line 158-176), along with an analysis of its differences from operational products. The 2B-CLDCLASS-LIDAR product, derived from the active remote sensing instrument CPR-CALIOP, is currently the most accurate cloud fraction product. When evaluating the accuracy of cloud retrieval algorithms against operational products, using 2B-CLDCLASS-LIDAR as the reference value is the only choice.

All other comments are about LSTM.

**Refereer3 comments:**

The manuscript primarily discusses the method, but the abstract and the content focuses only on the results. There is insufficient explanation about the method (machine learning and correction), its characteristics, and why it leads to improvements.

Answer: LSTM has been replaced by Random Forest, and the basic concepts, hyperparameter selection, and the reasons for its suitability for this study have been added(line230-267). Machine learning algorithms are not the reason for improvement in results. The main factors leading to improved results are the selection of true values, the presence of strong relationships between inputs and outputs, that is, the accuracy of the training set. If the training set is sufficiently accurate, regardless of the machine learning algorithm used, the results will be similar.

The manuscript lacks a discussion about the impact of different machine learning structures on the

retrieval of cloud fraction. Please provide more results regarding this.

Answer: The manuscript title has been changed to "Retrieval of cloud fraction using random forest based on FY4A AGRI observations." No need to discuss the impact of different algorithms on the results.

The data resolution of CloudSat and CALIPSO is not consistent with AGRI. It is crucial to discuss this dataset uncertainty and its impact on the retrieval.

Answer: Indeed, the resolutions of CloudSat & CALIPSO are different from that of AGRI. Therefore, spatial matching is necessary when creating the training dataset, with specific details in lines 188 to 193.

The captions of the figures and tables are too simplified and do not provide essential information.

Answer: The titles of the figures and tables have been changed.

Line 63: What is the spatial resolution of the instrument?

Answer: 4km。The relevant information has been added in the respective locations. Line 98.

Lines 65-81: Summarize these old and classic references, emphasizing only the important ones.

Answer: A summary of these references has been made. Line 63-68.

Lines 98-113: The logic is not clear. Why did the authors conduct this work, and what is the advantage of the LSTM compared to other frameworks?

Answer: The logic has been clarified and this section has been rewritten. Line 96-115.

Table 1: A reference should be provided.

Answer: This table is from https://www.nsmc.org.cn/nsmc/cn/instrument/AGRI.html. Line 134.

Lines 149-155: Since cloud fraction is important for building the training dataset, the algorithm for the joint product should be introduced. Additionally, how the uncertainty in the dataset affects the retrieval should also be discussed.

Answer: The algorithm for the joint products is detailed. Line 158-176.

Lines 253-258: When discussing LSTM, the use of time series information is implied. Is this the case in your study?

Answer: No time series information has been used.

Line 81, it should be made clear that why "season" and "climate" will influence the thresholds?

Answer: Line 70-79. The reason has been added.

All minor corrections have been rectified.

---

## Author Response (AR2)

In this version, the main modifications include: 1. The use of two algorithms, Multi-Layer Perceptron (MLP) and Random Forest (RF). Different combinations of hyperparameters were tested to analyze their impact on model performance, and the best combination was selected. 2. The time range of the dataset is from August 2018 to July 2019. The ratio of various cloud fraction in the training set is 0: 0.16: 0.33: 0.5: 0.76: 0.83: 1= 5: 1: 1: 1: 1: 1: 5. After these adjustments, the model's performance showed significant improvement.

Refereer3 unresolved previous comments:

The manuscript primarily discusses the method, but the abstract and the content focuses only on the results. There is insufficient explanation about the method (machine learning and correction), its characteristics, and why it leads to improvements.

Answer: Based on your feedback, a more detailed introduction to machine learning methods has been provided, along with a comparison of the impact of different methods on retrieval.

Line 237-356

The manuscript lacks a discussion about the impact of different machine learning structures on the retrieval of cloud fraction. Please provide more results regarding this.

Answer: Based on your feedback, we have added comparisons of different machine learning algorithms and tested various combinations of hyperparameters to assess their impact on machine learning performance.

The data resolution of CloudSat and CALIPSO is not consistent with AGRI. It is crucial to discuss this dataset uncertainty and its impact on the retrieval.

Answer: The data resolution of CloudSat and CALIPSO is not consistent with that of AGRI, so when creating the training dataset, it is necessary to perform spatiotemporal matching of the data. Ensure that the time range of the matched data from the two instruments is within 15 minutes and the distance range is within 1.5 km. Additionally, within the AGRI pixel, at least two CloudSat and CALIPSO pixels should be covered. After matching, the cloud fraction detected by CloudSat and CALIPSO can better represent the actual cloud fraction within the AGRI pixel.

However, the errors in the matched dataset are unavoidable. The AGRI scanning method operates from left to right and top to bottom. Each complete scan of the full disk takes 15 minutes and generates a dataset. It is impossible to determine the exact moment of a specific point within the full disk. This limits the time range for matching datasets to within 15 minutes. However, in areas with higher wind speeds, clouds can move a significant distance within that 15-minute window. Therefore, errors arising from timing issues cannot be avoided.

Line 187-199

Refereer2 comments:

Given these results, I think that readers will need to be convinced that you have selected a reasonable algorithm with reasonable hyperparameters. Given the wide difference in results between the

two algorithms so far, I don't think you are reaching the ceiling of possible model skill given your data.

• I think you need to experiment with more hyperparameters for the random forest (and report those experiments here) and/or compare with some other baseline algorithms with reasonable hyperparameter choices.

• Given that your RF does not consistently outperform a 3-hidden-unit MLP (an incredibly small network), I'd be surprised if it significantly improves upon a linear regression model or a model that simply outputs the mean of the training data. These would be useful tests for you, and they would help to understand the behavior of your evaluation metrics.

• I expect you can achieve better performance with larger trees / forests, or with an MLP with more than 1 layer and more than 3 hidden units per layer. I think you should try both. Trying more hyperparameters for your RF implementation should be very simple.

• If you already have code that can train an LSTM on your data, it should not be too complicated to try something like a 3-layer MLP with, for example, 16 hidden units per layer. This is still a very simple network, but much more expressive than the 3-unit MLP you tried before. I also think a well-designed convolutional neural network would significantly outperform your current results.

Answer: Two algorithms, RF and MLP, were used, and through experimentation, the optimal hyperparameter combinations for both were determined.

In my previous review, I suggested you use data with broader seasonal coverage. You responded: "A single observation from FY4A AGRI, the northern and southern hemispheres contain data from different seasons, climates, and surface types. Therefore, the training dataset required for training the model does not need to cover a long period of time." I disagree. While your data covers both hemispheres, there is a relationship between latitude, seasonality, and clouds. I see no reason not to use data from more times of year. If there is too much raw data, you can randomly sample the same amount of data you currently have, but from more seasons / years.

Answer: The time range of the dataset has been extended to one year, with 80% allocated for training and 20% for testing.

Line 224-232

Finally, you need to discuss the numbers achieved by other methods which attempt the same or similar tasks. For cloud detection, your highest recall (POD) is about 0.9, and your highest precision (1 – FAR) is about 0.82, but you also get values as low as 0.67. This seems low to me, especially when these numbers aren't far from multilayer cloud detection numbers using similar inputs and 2B-CLDCLASS-LIDAR labels (Ding et al 2022. Multilayer detection is a (much) harder task. Still, it's hard to compare, as you only report precision/recall (POD / FAR) while other papers seem to report accuracy. You should include accuracy alongside POD / FAR in your results. As for the ME / RMSE values, these seem high to me. ME is a little misleading when compared to RMSE, as your positive / negative errors cancel out, but the values are still quite high. Just because your method outperforms the operational product does not mean it is competitive, it just means the operational product is bad. The results (especially in light of the inconsistency between the MLP results and the RF results) suggest to me that you have more work to do when it comes to selecting and training the model.

Answer: For the characterization of cloud detection accuracy, it has been changed to directly display the confusion matrix (as shown in Figure 1 of the manuscript). Figure 1 includes recall rate, false alarm rate, and accuracy. The values for ME and RMSE have significantly decreased. The value of ME can be misleading due to cancellation of positive and negative values, so the Mean Absolute Error (MAE) has been added.

Line-by-line comments:

130: I can't find Qu et. al. in your citations. I stumbled upon this, and I have not checked all other citations.

Answer: I have added this reference to the citation.

139: I tried to look up the Hu et. al. paper, but the DOI links to a different author / title than the one you list.

Answer: doi:10.27248/d.cnki.gnjqc.2020.000625

307: You cite Quesada-Ruiz et. al. 2022, which is a very specific type of random forest algorithm called AFGRRF. Do you use their method, or a more general random forest algorithm? If it is the former case, you need to explicitly state that you use this method as well as briefly summarize the method (here you are only summarizing the general random forest algorithm). If instead you use a more general random forest, you should probably cite a seminal random forest paper, a survey paper, and/or a paper that applies (regular) random forests to similar data. I've noticed that Quesada-Ruiz et. al. have a publicly available implementation of AFGRRF in R. There are plenty of other publicly available implementations of random forests, so I'm curious why you use this method. It is important that you justify the choice of method in the text.

Answer: I used a more general random forest model. I have already made changes to the citations in the manuscript.

Line 254

316-332: Citations are needed throughout this section to back up your claims. This is especially true in the absence of experiments showing that your hyperparameter choices are competitive.

323-325: If you're using sqrt(M) for your Mtry parameter, shouldn't you be using 4 and 3 instead of 3 and 2? Sqrt(14) is closer to 4 than to 3, and sqrt(8) is closer to 3 than to 2.

Answer: This part of the content has been changed.

324: Which channels are not available at night? You should include this in Table 1.

Answer: The first six visible light channels have no values at night, meaning that channels with a central wavelength less than or equal to 2.225 are unavailable during nighttime.

Line 131-133

---

## Author Response (AR3)

We are very grateful to the reviewers' critical comments and thoughtful suggestions. Based on these comments and suggestions, we have made careful modification on the original manuscript. we acknowledge your comments and constructive suggestions very much, which are valuable in improving the quality of our manuscript. Here are our responses to the reviewers' comments one-by-one.

*Overall comments:*
*I am starting to feel confident that I mostly understand your method. Still, it has taken me several re-reads of the paper. Most of your readers will not take the time to understand it as well as your reviewers do. Some simple changes will make the paper much easier to follow:*
*• You should summarize what the inputs / outputs of your cloud detection and cloud fraction models are. If you can make this into a figure / flowchart, that's even better. You should include the actual shapes of those inputs / outputs.*

Anwser: The flow chart has been added as required. Line 245.

*• You should consolidate some of your results tables to highlight the comparison between your method and the operational approach.*

Answer: Table 2 and Table 3 are merged tables. Line 336, 349.

Line-by-line comments:

*210: Instead of "A dataset .. and B dataset" it is clearer to say "dataset A… and dataset B"*

Answer: Corrected.

*218: "To ensure the balance and representativeness of the samples, the proportions of different cloud fraction samples in dataset A are set at 5:1:1:1:1:1:5"218:*
*This makes sense. It would be interesting to see a comparison with the results of your previous approach (the 1:1:1:1:1:1:1 dataset), perhaps in the supplementary material.*

Answer: Perhaps I didn't explain the previous method clearly either. The previous method is 1:1:1, that is, the different cloud fraction in partly cloudy are uncertain. The results of the previous method are as follows.

| | | 5:1:1:1:1:1:5 | | | | 1:1:1 | | | |
|---|---|---|---|---|---|---|---|---|---|
| | Sky Classification | Day RF | Night RF | Day MLP | Night MLP | Day RF | Night RF | Day MLP | Night MLP |
| POD | Clear Sky | 0.964 | 0.919 | 0.959 | 0.905 | 0.935 | 0.895 | 0.931 | 0.890 |
| | Partly cloudy | 0.914 | 0.845 | 0.895 | 0.808 | 0.784 | 0.730 | 0.752 | 0.695 |

| | | | | | | | | |
|---|---|---|---|---|---|---|---|---|
| | Overcast | 0.959 | 0.919 | 0.957 | 0.920 | 0.926 | 0.910 | 0.924 | 0.904 |
| | Clear Sky | 0.047 | 0.102 | 0.064 | 0.131 | 0.128 | 0.162 | 0.157 | 0.193 |
| FAR | Partly cloudy | 0.078 | 0.153 | 0.085 | 0.172 | 0.152 | 0.215 | 0.159 | 0.225 |
| | Overcast | 0.038 | 0.061 | 0.039 | 0.063 | 0.077 | 0.089 | 0.078 | 0.098 |

[Figure]

Figure 1: In the training sample set, clear sky: partly cloudy: overcast = 1:1:1. That is, the accuracy of each model when the proportion of cloud fraction in partly cloudy is unknown.

(a) (b)

[Figure]

[Figure]

Figure 2: The accuracy of each model when 0:0.16:0.33:0.5:0.67:0.83:1 = 5:1:1:1:1:1:5 in the training sample.

• *I'm still a little confused about dataset B. In line 190 you say "at least two 2B-CLDCLASS-LIDAR pixels are required within each AGRI field of view. The cloud fraction average of these pixels is used as the cloud fraction for that AGRI pixel." Are you sampling so that the dataset approximates the 5:1:1:1:1:1:5 ratio? Also, if the labels are averaged between multiple 2B-CLDCLASS-LIDAR pixels, having a discretization like this doesn't make much sense.*

Answer: At least two 2B-CLDCLASS-LIDAR pixels are required within each AGRI field of view. The average cloud fraction of these pixels is used as the cloud fraction of this AGRI pixel. However, I found that the cloud fractions of 2B-CLDCLASS-LIDAR pixels within the AGRI field of view are mostly the same. After averaging, the proportions of cloud fractions of [0.16, 0.33, 0.5, 0.67, 0.83] are extremely high. Therefore, I ignored other cloud fraction situations with extremely small proportions. Doing so can also better balance the training samples. The following is the number of occurrences of different cloud fractions after averaging in two daytime samples that I counted. Line 213-219.

[Figure]

*265: You say MLP uses stochastic gradient descent, but you later report that your solver is Adam. These are different optimizers. You can just say "the model's weights are trained in a supervised manner using backpropagation."*

Answer: Revised as per the suggestion.

*265: What loss function do you use to train the cloud detection and cloud fraction MLPs? I would assume cross-entropy for the cloud detection model, and MSE for the cloud fraction model, but you need to mention it.*

Answer: For the loss function, the cloud detection model is cross-entropy, and the cloud score model is MSE. It has been mentioned at the corresponding position in the text. Line 272-274.

*278: "Hidden layer size… Hidden layer neuron count"*
*Hidden layer size and hidden layer neuron count are the same thing. This explains why in the previous version of your paper I believed your MLP had only 3 neurons. You should instead use "number of hidden layers" to describe what you are currently calling "hidden layer size" and you can use "hidden layer size" to describe what you are currently calling "hidden layer neuron count." The term "neuron" has become outdated as MLPs / neural networks have drifted away from their original neuroscientific inspiration, and if you need to refer to a single node in a hidden layer*

*you can call it a "hidden unit.*

Answer: Corrected.

*277-288:*
*Getting results with different numbers of layers and different hidden layer sizes is great. Still, it would be nice to see the actual results. A few notes on this section:*

*• This should be a table. It is difficult to read at present.*

Answer: Sorry, I didn't add a table here. After unifying the batch size, the only parameter that is different for each model is the number of hidden layers. One variable is not suitable for making a table.

*• Since you have all of these results, you should put them in the supplementary material, so interested readers can benefit from your experiments for their own work.*

Answer: The following are the results obtained by using different numbers of hidden layers.

Table 1. The influence of different numbers of hidden layers on the accuracy of the models.

| number of hidden layers | 2 | 3 | 4 | 5 | 6 | 7 | 8 | 9 |
|---|---|---|---|---|---|---|---|---|
| Day CL accuracy | 0.9067 | 0.9122 | 0.9337 | **0.9369** | 0.9360 | 0.9355 | 0.9355 | 0.9364 |
| Night CL accuracy | 0.8605 | 0.8691 | 0.8838 | **0.8878** | 0.8843 | 0.8795 | 0.8845 | 0.8849 |

Table 2. The influence of different numbers of hidden layers on the precision of the models.

| number of hidden layers | | 2 | 3 | 4 | 5 | 6 | 7 | 8 | 9 |
|---|---|---|---|---|---|---|---|---|---|
| Day RE | ME | -0.0047 | 0.0037 | **-0.0009** | -0.0101 | -0.0024 | 0.0044 | 0.0042 | -0.0009 |
| | MAE | 0.1397 | 0.1240 | **0.1053** | 0.1048 | 0.1065 | 0.1036 | 0.1032 | 0.1301 |
| | RMSE | 0.1677 | 0.1513 | **0.1332** | 0.1334 | 0.1314 | 0.1312 | 0.1319 | 0.1303 |
| Night RE | ME | -0.0006 | --0.0059 | 0.0009 | 0.0070 | **-0.0032** | -0.0006 | -0.0062 | -0.0043 |
| | MAE | 0.1613 | 0.1510 | 0.1413 | 0.1371 | **0.1322** | 0.1325 | 0.1310 | 0.1321 |
| | RMSE | 0.2133 | 0.1810 | 0.1716 | 0.1630 | **0.1623** | 0.1633 | 0.1665 | 0.1625 |

According to the results in the above table, the following conclusions can be drawn: (1) MLP classification model for daytime: number of hidden layers = 5. (2) MLP classification model for nighttime: number of hidden layers = 5. (3) MLP regression model for daytime: number of hidden layers = 4. (4) MLP regression model for nighttime: number of hidden layers = 6.

*• You don't need to try so many batch sizes. I would just pick one and keep it consistent*

*across all experiments. Usually it's a reasonable choice to pick the biggest batch size you can fit into your GPU's VRAM.*

Answer: The batch size of all models is unified as 1500.

*• You don't need to report results with different activation functions. It is extremely well established in the literature at this point that ReLU and ReLU variants are almost always the best choice.*

Answer: Corrected.

*• You should use the terms themselves and explain what they mean rather than listing variable names (e.g. LearnRateDropPeriod).*

Answer: Corrected.

*311-474: You should find a way to combine the operational product tables with the tables of your method. Directly comparing them will highlight the whole point of your paper.*

Answer: Merged.

*414: There is a lot of white space in this and other figures. Trimming some white space would make the panels bigger and more legible.*
*Panel (b):*
*• "y = 0.8092 + 0.2441": missing an "x"*
*• I'm confused by these results. Line 205 reads: "the cloud fraction is the average of cloud fractions at different layers." Why are the true cloud fraction values in this plot discretized to [0, 0.16, 0.33, ...]? Isn't your model predicting an average? This is related to my above comment about line 218.*

Answer: Sorry, I didn't express clearly here.

     1) Cloud fraction is the average of cloud fractions of different layers: In the sample set of the sun glint area, only two situations occur, namely one-layer cloud and two-layer cloud. When there are two layers of cloud, there is always one layer with a cloud fraction of 1. According to the previous description, when there is one layer with a cloud fraction of 1, this pixel should be regarded as overcast. Some data is shown in the figure below.

     2) The average of cloud fractions of at least two pixels: Due to the very small area of the sun glint area, it is very difficult to match. If at least two CloudSat & CALIPSO pixels within an AGRI pixel are required, this will make the available sample size very small. Therefore, when making the sample set of the sun glint area, only one CloudSat & CALIPSO pixel within an AGRI pixel is required. The above two points are the reasons why the true value is discrete points. Line 374-386.

| | 1 | 2 | 3 | 4 |
|---|---|---|---|---|
| 58 | 1 | 1 | -99 | -99 |
| 59 | 0.1667 | 1 | -99 | -99 |
| 60 | 1 | 1 | -99 | -99 |
| 61 | 0.3333 | 1 | -99 | -99 |
| 62 | 0.3333 | 1 | -99 | -99 |
| 63 | 1 | 1 | -99 | -99 |
| 64 | 0.1667 | 1 | -99 | -99 |
| 65 | 0.6667 | 1 | -99 | -99 |
| 66 | 0.8333 | 1 | -99 | -99 |
| 67 | 0.6667 | 1 | -99 | -99 |
| 68 | 0.3333 | 1 | -99 | -99 |
| 69 | 1 | 1 | -99 | -99 |
| 70 | 1 | 1 | -99 | -99 |
| 71 | 0.1667 | 1 | -99 | -99 |
| 72 | 0.3333 | 1 | -99 | -99 |
| 73 | 1 | -99 | -99 | -99 |
| 74 | 0.1667 | 1 | -99 | -99 |
| 75 | 0.1667 | 1 | -99 | -99 |
| 76 | 1 | -99 | -99 | -99 |
| 77 | 1 | -99 | -99 | -99 |
| 78 | 1 | -99 | -99 | -99 |

*459: Panel (a) should have a color bar, like the other panels.*
*488: Panel (a) should have a color bar. Panel (c) has a color bar that is difficult to read. It should be consistent with panel (b). Also you could trim some white space in this figure, and make it large enough to fill the page width (like Figure 3).*

Answer: Corrected.

*497: "CloudSat" not "Cloudsat", and "CALIPSO" not "Calypso"*

Answer: Corrected.

*References: I still cannot find the Hu, J. paper anywhere on the internet, with either the DOI or the title. Please make sure all citations have a verifiable link or DOI.*

Answer: I have replaced this references.

[1] Chai D ,Huang J ,Wu M , et al.Remote sensing image cloud detection using a shallow convolutional neural network[J].ISPRS Journal of Photogrammetry and Remote Sensing,2024,20966-84.